# Annealing Condition Effects on the Structural Properties of FePt Nanoparticles Embedded in MgO via Pulsed Laser Deposition

**DOI:** 10.3390/nano11010131

**Published:** 2021-01-08

**Authors:** Tingting Xiao, Qi Yang, Jian Yu, Zhengwei Xiong, Weidong Wu

**Affiliations:** 1Science and Technology on Plasma Physics Laboratory, Research Center of Laser Fusion, CAEP, Mianyang 621900, China; xiaotingting@caep.cn (T.X.); yangqi807@caep.cn (Q.Y.); yujianroy@163.com (J.Y.); 2Joint Laboratory for Extreme Conditions Matter Properties, Southwest University of Science and Technology, Mianyang 621010, China; zw-xiong@swust.ed

**Keywords:** *L*1_0_ phase, FePt, phase transformation, annealing condition

## Abstract

FePt nanoparticles (NPs) were embedded into a single-crystal MgO host by pulsed laser deposition (PLD). It was found that its phase, microstructures and physical properties were strongly dependent on annealing conditions. Annealing induced a remarkable morphology variation in order to decrease its total free energy. H_2_/Ar (95% Ar + 5% H_2_) significantly improved the *L*1_0_ ordering of FePt NPs, making magnetic coercivity reach 37 KOe at room temperature. However, the samples annealing at H_2_/Ar, O_2_, and vacuum all showed the presence of iron oxide even with the coverage of MgO. MgO matrix could restrain the particles’ coalescence effectively but can hardly avoid the oxidation of Fe since it is extremely sensitive to oxygen under the high-temperature annealing process. This study demonstrated that it is essential to anneal FePt in a high-purity reducing or ultra-high vacuum atmosphere in order to eliminate the influence of oxygen.

## 1. Introduction

The FePt alloy attracts great research interest because of its potential in the ultra-high-density data magnetic storage [1,2,3,4], biological imaging [5], and enhanced catalyst for oxygen reduction reaction [6]. FePt can be produced with two phases (*A*_1_ and *L*1_0_ phase) when their atom ratio is close to 1:1. In *A*_1_ phase (with face-centered-cubic, *fcc* structure), Fe and Pt randomly occupy the site of its *fcc* lattice, which results in super para-magnetism in FePt at room temperature. In contrast, its *L*1_0_ phase with face-centered tetragonal (*fct)* structure presents alternating layers of Fe and Pt atoms in the (002) planes and tetragonal distortion (c/a = 0.96). The ordered *L*1_0_-FePt has a large uni-axial magneto crystalline anisotropy (*K*u = 7 × 10^6^ J/m^3^), which contributes to the thermal stability and excellent ferromagnetic properties in FePt even with size down to ~3 nm [7]. Since the synthesized FePt nanoparticles always show chemically disordered *A*_1_ phase, a thermal annealing process is necessary to convert this disordered *fcc* structure to an ordered *fct* structure. Thus, the post-annealing procedure is a key step for FePt to obtain good ferromagnetism properties. However, a high-temperature post-annealing process will lead to coalescence of NPs and result in magnetic coupling between those ferromagnetic clusters. One way of solving this problem is to embed FePt NPs in a nonmagnetic oxide matrix such as SiO_2_ [8,9,10,11], Al_2_O_3_ [7], MgO [12,13], and NaCl [14,15,16,17]. Matrix material could offer a homogenous single crystal environment to FePt, and control the orientation of its easily magnetized axis.

Further, many works have been done to investigate the effect of annealing conditions on the structure and magnetic properties of FePt. Most of them just focused on annealing temperature, time, and heating rate [4,18,19,20,21,22,23]. The annealing atmosphere (i.e., vacuum, Ar, N_2_, or forming gases, etc.) has a pronounced impact on the structural transition and related magnetic properties for FePt. In previous contributions, it has been shown that highly coercive FePt films can be obtained by annealing in hydrogen rather than in a vacuum [24,25,26,27]. It has also been reported that nitrogen corporation forming FeN enhances the diffusivity of Fe and Pt during annealing and thus improves the *L*1_0_ ordering [28]. Moreover, some previous studies claimed that MgO matrix can act as protection of FePt NPs from external oxidation [29,30]. Hsiao et al. report that no oxidation of Fe occurred after annealing at 800 °C under a poor vacuum condition (~10^−4^ Pa) [31], while some other researchers reported severe oxidation of FePt which was covered with Al_2_O_3_ [32,33] or MgO [34]. In order to figure out the conflict, it is essential to study the annealing reaction under different conditions to further understand the annealing process of FePt NPs with the coverage of matrix materials.

In this work, FePt NPs were embedded into single-crystal MgO by pulsed laser deposition (PLD). The samples were annealed at different temperatures and gas environment. The annealing mechanisms and micro-structural features of those samples with different annealing conditions were discussed, together with their magnetic properties.

## 2. Experimental Details

The FePt-MgO nanocomposite films were grown by PLD on MgO (001) single-crystal substrates. The background pressure and deposition pressure are ~3.0 × 10^−6^ Pa and ~5.0 × 10^−5^ Pa, respectively. During the growth process, the pulsed laser with the wavelength 248 nm, pulsed frequency 2 Hz, and energy density 2.5 J/cm^2^ irradiated the Fe_50_Pt_50_ alloy target and pure MgO target alternatively, and the substrate temperature is maintained at 650 °C. Firstly, an MgO buffer layer was grown prior to the growth of the FePt layer in order to improve the surface quality of MgO substrate. After the deposition of the FePt layer, a continuing MgO cover layer with a thickness around 10 nm was deposited further in order to prevent FePt NPs from exposure to air. After the completion of every MgO layer, the sample was exposed to oxygen atmosphere at 5 Pa for 20 min in order to eliminate the oxygen vacancies in MgO.

After the growth process, five samples with different annealing conditions were named as sample S1, S2, S3, S4, and S5, respectively. Detailed samples information is listed in Table 1. The compositions of samples were determined by energy-dispersive x-ray spectroscopy (EDS, EDAX Inc., Mahwah, NJ, USA). Structural investigations of these specimens were performed with high-resolution transmission electron microscopy (HRTEM) and high-resolution high-angle annular dark field scanning transmission electron microscopy (HAADF-STEM) by using a FEI Tecnai F20 microscope (FEI Inc., Hillsboro, OR, USA). To analyze the chemical states of Fe and Pt atoms, x-ray photoelectron spectroscopy (XPS, Thermo Fisher Scientific Inc., Massachusetts, MA, USA) was used to measure Mg *Kα* twin sources. In-plane and out-of-plane magnetization loops were measured at room temperature with superconducting quantum interference device magnetometer (SQUID, Quantum Design Inc., San Diego, CA, USA).

## 3. Results and Discussion

The composition of the samples was determined to be 52:48 by EDS. Top-view TEM images of S1 and S2 are shown in Figure 1 and a remarkable morphology change was observed. For S1, the FePt layer shows an interconnected isotropic maze-like pattern and almost covered all the MgO substrate. However, the atoms’ diffusion occurred, and film morphology changed from interconnected to granular after annealing. For S2, the particle size is around 5~13 nm, and their distance is approximately 10 nm. This morphology variation after annealing could be explained by de-wetting phenomena. Many earlier works have also reported this similar phenomenon about metal (like Au, Pt, Co-Pt, etc.) films after thermal treatment [35,36]. Here, de-wetting phenomena occurred in order to decrease its total free energy, which caused the reduction of interfacial area between the FePt layer and substrate [36]. Therefore, the diffusion of FePt atoms under a high temperature prefers to assemble at the top of FePt in order to decrease their interfacial area. It drove the edge retraction of the FePt layer and changed it towards a collection of three-dimensional (3D) islands. It demonstrates that the coverage of the MgO matrix could restrain the particles’ coalescence effectively, but could not disturb the diffusion and island growth of FePt atoms.

Further, the influence of annealing temperature on particle morphology was also investigated by using TEM. Figure 2 gives the top and cross-section HRTEM images of S2 and S3. It can be seen that the thickness of the MgO cover layer is around 10 nm. The stripe contrast observed in the particles is Moiré pattern caused by the lattice parameter difference between MgO matrix and FePt NPs. The HRTEM images show that the crystal structure of the FePt NPs in both samples remains disordered *A*_1_ phase, but their morphology varies with annealing temperature. As shown in Figure 2a,b, the FePt NPs annealed at 800 degrees (S2) exhibit a well-defined morphology with facets. High-resolution TEM observation of FePt NPs (Figure 2b) indicates very good crystalline and a dominant truncated inverted triangle shape. The formation of these shapes may result from the anisotropy of FePt NPs surface energy and the sufficient atoms’ diffusion in higher temperatures [37]. While, as shown in Figure 2c,d, the FePt NPs in S3 exhibit a sphere-like morphology in both top and cross-section view images. Further, the elemental distribution of S2 within the orange area was examined by using EDS mapping, as shown in Figure 2e. It reveals the projected distributions of Pt and Fe within the particle. Figure 2e insert shows the electron energy loss spectroscopy (EELS) spectrum of S2, with its Fe2p_3/2_ and Fe2p_1/2_ peaks located at 708.8 and 721.6 eV, respectively. These values are higher than the ones reported for bulk values of pure Fe (Fe (2p_3/2_) = 706.75 eV, Fe (2p_1/2_) = 719.95 eV) [38]. It suggests that the Fe atoms of S2 were oxidized after annealing. It also demonstrates that the coverage of MgO cannot avoid the oxidation of Fe when it is exposed to oxygen in high temperatures.

Figure 3 compares the x-rays diffraction (XRD) spectra of the sample before annealing and the samples annealed at 800 degrees for 4 h, but with different annealing atmosphere. All FePt-MgO nanocomposite films exhibit a sharp peak at 40.1°, which corresponds to FePt (111) texture. For S5, (001) and (002) peaks can be clearly observed, which indicate the existence of ordered *L*1_0_ phase, while for S2 and S4, there are no peaks corresponding to *L*1_0_-FePt. It suggests that no phase transition occurred under oxygen and high-vacuum annealing atmosphere. For S2, the oxidation of Fe may stop the occurrence of phase transition. But for S2 and S4, no peak can be identified with iron oxide. In order to figure out what stopped the phase transition in S4, a further study has been conducted by XPS.

Figure 4 illustrates the x-ray photoelectron spectroscopy (XPS) spectra of the Fe2p binding energy of the samples before annealing and the samples annealed at 800 degrees for 4 h but with different annealing atmosphere. A Shirley background was removed from the original spectra. For S1, two main peaks located at around 706.72 and 719.81 eV are assigned to Fe (0) Fe2p_3/2_ and Fe2p_1/2_, respectively. While for S2 and S4, both their main peaks are located at around 708.89 and 722.45 eV, which are assigned to iron oxide. It reveals that the oxidation of Fe atoms occurred under vacuum environment since oxygen content is not low enough at pressure around 5 × 10^−5^ Pa. While for S1 and S5, both the Fe2p_3/2_ and Fe2p_1/2_ peaks were composed of two peaks located at 706, 708.1 and 720, 722 eV, respectively. The two components for each peak were attributed to the coexistence of Fe (0) and iron oxide. It suggests that iron atoms in S1 and S5 were partially oxidized. The Fe oxide of S1 may result from the oxygen treatment process in order to eliminate the oxygen vacancies in as-grown MgO. It reveals that iron atoms are extremely sensitive to oxygen under high temperatures, even with the coverage of MgO. For S5, its relative intensity of oxidized component is higher than S1. It suggests that oxidized Fe has also been introduced from the annealing process, excepting from the oxygen treatment process. Ying [20] and Salahpour [39] reported similar oxidation phenomena of FePt, in which FePt was annealed in a forming gas (85% N_2_ + 15% H_2_ or 90% Ar + 10% H_2_), because oxygen content cannot be completely eliminated from the annealing furnace with flowing H_2_/Ar (N_2_). Hydrogen could suppress the oxidation process but could not effectively reduce oxidized Fe, and even traces of O_2_ present in the flowing H_2_/Ar mixture can cause oxidation. Further study on annealing temperature and time influence on its oxidation process will be conducted in our future work.

Figure 5a,b shows the STEM image and selected area electron diffraction of S5. S5 exhibits a similar micro-morphology to S2. As shown in Figure 5b, the reflections corresponding to two different compounds are clearly distinguished (FePt reflections are underlined). Beam direction is along the [001] zone axis of the substrate. The orientation relationship between FePt and MgO is < 100 > _FePt_ || <100 > _MgO_, {001} _FePt_ || {001} _MgO_. The strong {110} super lattice reflections indicate that FePt NPs with crystallographic c-axis oriented normal to the substrate surface with a high degree of order after annealing in H_2_/Ar atmosphere. Except for suppressing oxidation, Vladymyrskyi et al. reported that hydrogen plays an important role in the FePt ordering process [25]. The fast diffusion rate of hydrogen atoms in turn could induce an increased diffusion rate of Fe and Pt atoms and thus promote the *L*1_0_ ordering. Figure 5c shows a HAADF-STEM image of a typical FePt nanoparticle for S5. This particle exhibits *fct* structure and its (110) d-spacing is around 0.2712 nm. Furthermore, as marked with red circles and yellow circles, a few areas inside and in the edges of the particle present *A*_1_ phase. This nano-sized region with disordered feature in the edge may be caused by the tensile strain arising from the lattice misfit between FePt and MgO, while the disorder region inside the particle may be caused by insufficient annealing.

Figure 6a shows the out-of-plane magnetization curves obtained from the samples with different annealing conditions. Except for S5, all the other samples showed a completely soft ferromagnetic behavior, which is for the *fcc* disordered phase. However, after a heat treatment under H_2_/Ar atmosphere, the specimen exhibits a big coercivity, which indicates a transition from soft to hard ferromagnetic phase. As shown in Figure 6b, the in-plane and out-of-plane coercivity of S5 were found to be 0.9 and 37 KOe, respectively. The great disparity between in-plane and out-of-plane coercivity indicates that the easy axes of the magnetic grains were perpendicular to the substrate. Moreover, the presence of a shoulder at low applied magnetic field in the hysteresis was caused by the presence of soft phase due to the partial ordered transformation [29,40]. The soft phase resulted from the element segregation and oxidation of Fe, as mentioned from XPS and TEM analyses above.

## 4. Conclusions

In this work, series of FePt-MgO nanocomposite films were successfully fabricated and the samples were annealed at different temperatures and gas environment. Annealing induced a remarkable morphology variation and higher temperature could result in a well-defined morphology with facets. The samples annealing in oxygen, vacuum, and H_2_/Ar atmosphere all show the presence of iron oxidation, which suggests that the iron is extremely sensitive to oxygen under the high-temperature annealing procedure even with the coverage of MgO. Annealing in reducing atmosphere (H_2_/Ar) suppressed the oxidation process and the specimen has a magnetic coercivity of 37 KOe at room temperature. Therefore, the present work demonstrates that the oxygen and hydrogen content play an important role in controlling the microstructure and magnetic properties of FePt-MgO nanocomposite films under the post-annealing procedure. It is essential to anneal FePt in a high-purity reducing or ultra-high vacuum atmosphere in order to eliminate the influence of oxygen.

## Figures and Tables

**Figure 1 nanomaterials-11-00131-f001:**
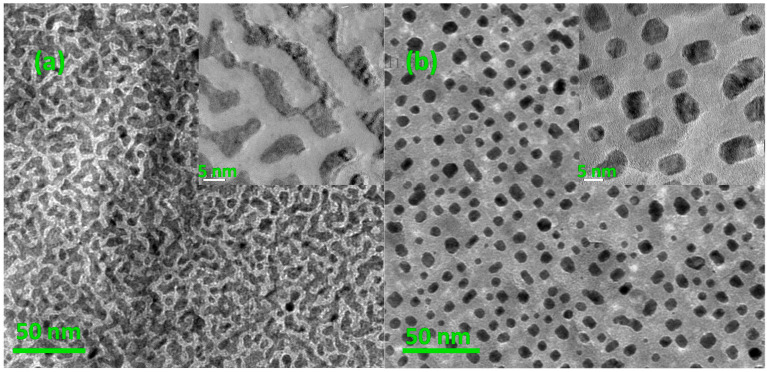
Top-view Transmission Electron Microscopy (TEM) images of (**a**) S1: before annealing (insert is its local magnification image) and (**b**) S2: after annealing in oxygen at 800 °C for 4 h (insert is its local magnification image).

**Figure 2 nanomaterials-11-00131-f002:**
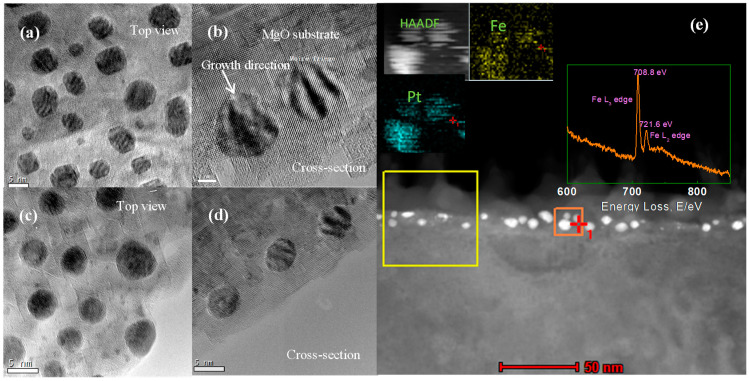
(**a**,**b**) Top and cross section HRTEM views of S2 (annealed under oxygen at 800 °C for 4 h). (**c**,**d**) Top and cross-section HRTEM views of S3. (**e**) The cross-section STEM image of S2 (annealed under oxygen at 700 °C for 4 h). Insert: EELS spectrum of yellow square area and EDS mapping of orange square area.

**Figure 3 nanomaterials-11-00131-f003:**
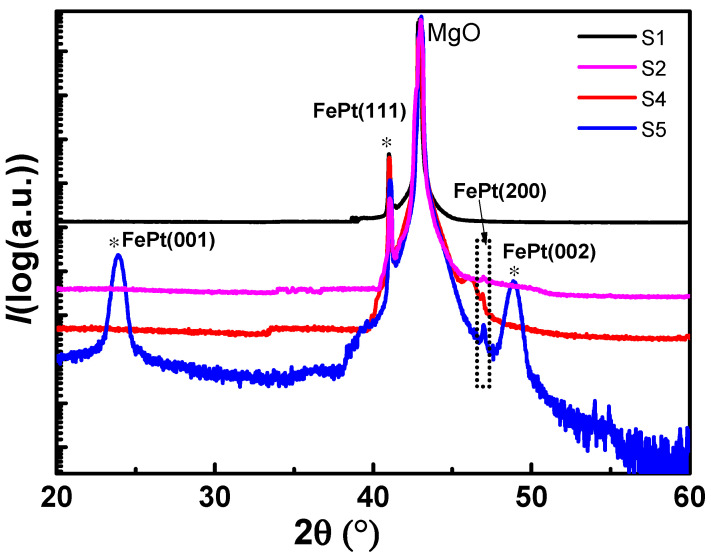
XRD spectra for the samples with different annealing conditions.

**Figure 4 nanomaterials-11-00131-f004:**
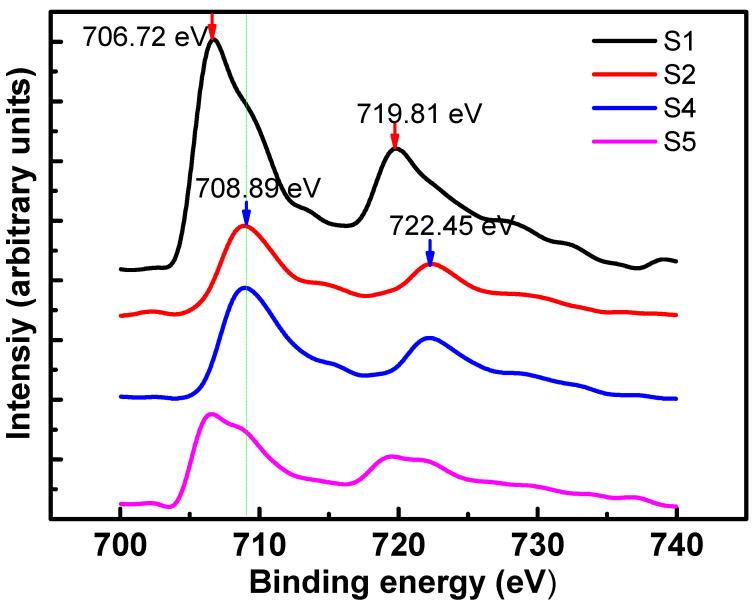
XPS spectra of the Fe2p binding energy of the samples with different annealing conditions.

**Figure 5 nanomaterials-11-00131-f005:**
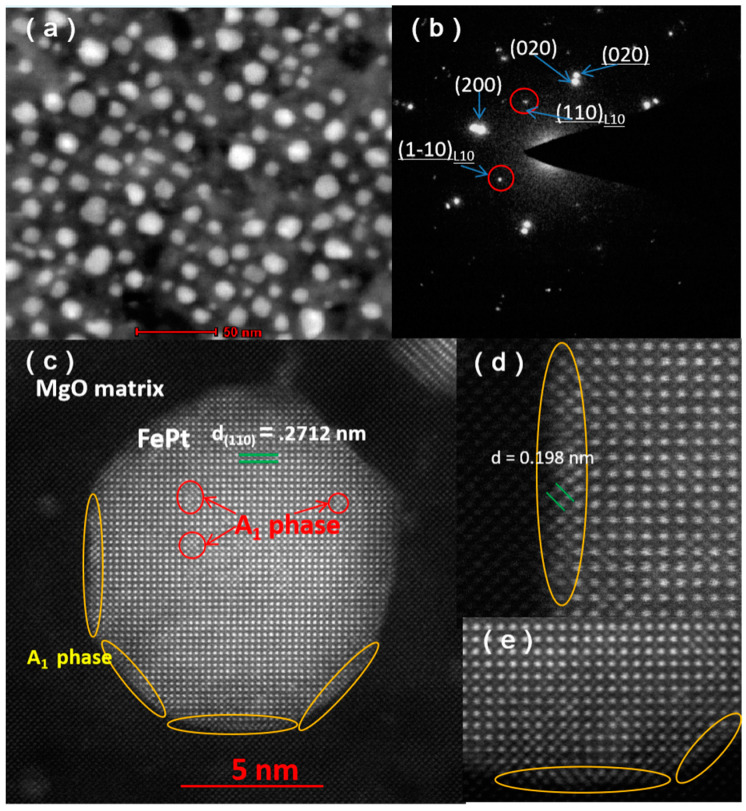
HAADF-STEM image (**a**) and selected area electron diffraction (**b**) of S5, (**c**) HAADF-STEM image of a typical FePt nanoparticle for S5, (**d**,**e**) enlarged HAADF-STEM images of the particle.

**Figure 6 nanomaterials-11-00131-f006:**
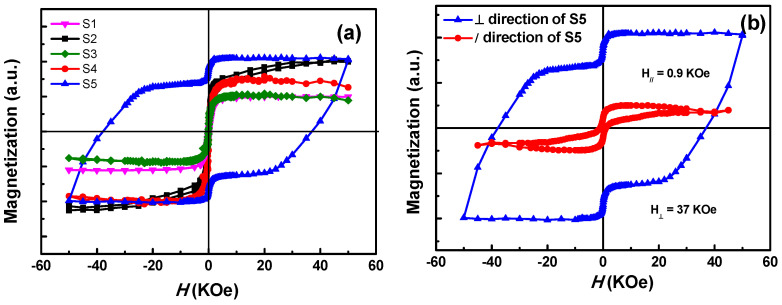
(**a**) Magnetization curves of samples with different annealing conditions, (**b**) in-plane and out-of-plane magnetization curves of S5.

**Table 1 nanomaterials-11-00131-t001:** Samples information.

Sample No.	Atmosphere	Temperature (°C)	Time (Hours)
S1	unannealed	--	--
S2	O_2_ (~5 Pa)	800 °C	4
S3	O_2_ (~5 Pa)	700 °C	4
S4	high vacuum (~5 × 10^−5^ Pa)	800 °C	4
S5	H_2_/Ar (95%Ar + 5% H_2_)	800 °C	4

## Data Availability

Data available in a publicly accessible repository that does not issue DOIs.

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
