# Peer review of "Annealing Condition Effects on the Structural Properties of FePt Nanoparticles Embedded in MgO via Pulsed Laser Deposition"

_nanomaterials, 2021, doi:10.3390/nano11010131_

Round 1
Reviewer 1 Report
This is an interesting paper. The present work demonstrates that the oxygen and hydrogen content play an important role in controlling the microstructure and magnetic properties of FePt-MgO nanocomposite films under post-annealing treatments. The authors suggest that it is essential to anneal FePt in a high purity reducing or ultra-high vacuum atmosphere in order to eliminate the influence of oxygen. The reviewer believes that this work is worth publishing. The English is problematic and needs to be reviewed by a person very proficient in the English language. This is a significant problem.
Author Response
Reviewer 1: This is an interesting paper. The present work demonstrates that the oxygen and hydrogen content play an important role in controlling the microstructure and magnetic properties of FePt-MgO nanocomposite films under post-annealing treatments. The authors suggest that it is essential to anneal FePt in a high purity reducing or ultra-high vacuum atmosphere in order to eliminate the influence of oxygen. The reviewer believes that this work is worth publishing. The English is problematic and needs to be reviewed by a person very proficient in the English language. This is a significant problem.
Response:Thanks very much for your suggestion and positive comments. After carefully check, we found many grammar and sentence errors, and have modified the manuscript accordingly. We hope the revised paper will be more clear and accurate on expressions.
Reviewer 2 Report
Review of the manuscript entitled "Annealing Condition Effects on the Structural Properties of FePt Nanoparticles Embedded in MgO via Pulsed Laser Deposition" presented by Weidong Wu and co-workers. The paper considers issues of preparing and structural properties of the FePt alloy embedded in magnesium oxide (MgO) obtained via pulsed laser deposition (PLD).
First of all, the subject and the results are not new and improperly supported by experimental data and one can argue about the impact of these results in the field of alloy composites as well as high density magnetic recording media, especially their structural and magnetic properties. Carefully examination of the manuscript reveals that the Authors have published some similar papers and even some figures had been presented in the similar way, for example:
- Jian Yu, Tingting Xiao, Jin Wang, Yunhui Tang, Xuemin Wang, Bin Li, Weidong Wu and Yafei Zhang, The structural evolution in the growth process of FePt embedded in MgO matrix, J. Mater. Sci. (2020) 55:12305–12313.
- Jian Yu, Tingting Xiao, Xuemin Wang, Xiuwen Zhou, Xinming Wang, Liping Peng, Yan Zhao, Jin Wang, Jie Chen, Hongbu Yin, and Weidong Wu, A Controllability Investigation of Magnetic Properties for FePt Alloy Nanocomposite Thin Films, Nanomaterials, 2019, 9(1): 53.
Moreover, the Authors should clearly show here in what sense their materials are better than the others investigated earlier. As a conclusion, the topic could be interesting, due to the above criticisms I cannot recommend this paper to be published by Nanomaterials - an Open Access Journal from MDPI.
Author Response
Reviewer 2: Review of the manuscript entitled "Annealing Condition Effects on the Structural Properties of FePt Nanoparticles Embedded in MgO via Pulsed Laser Deposition" presented by Weidong Wu and co-workers. The paper considers issues of preparing and structural properties of the FePt alloy embedded in magnesium oxide (MgO) obtained via pulsed laser deposition (PLD).
Point 1: First of all, the subject and the results are not new and improperly supported by experimental data and one can argue about the impact of these results in the field of alloy composites as well as high density magnetic recording media, especially their structural and magnetic properties.
Response 1: Thanks for your comments. The subject of this work is meaningful and we believe that it could also inspire some researchers in related field. As mentioned in our revised introduction, large amount of papers claimed that they can obtain ordered FePt once it was annealed in vacuum (even poor vacuum) and the matrix material could protect FePt from oxidation effectively. But we always failed when we did the same experiment. Further investigation found that oxygen plays key role on the FePt transition process and it should be eliminate from its annealing atmosphere. Normally, people think 10 nm crystallized MgO could stop the diffusion of oxygen from MgO onto the surface of FePt nanoparticles, but our experiment show opposite result. There is big difference between previous reports and there is still many issue needs to be solved, so we believe that this work is still worth to publish and inspire readers.
Point 2: Carefully examination of the manuscript reveals that the Authors have published some similar papers and even some figures had been presented in the similar way, for example:
- Jian Yu, Tingting Xiao, Jin Wang, Yunhui Tang, Xuemin Wang, Bin Li, Weidong Wu and Yafei Zhang, The structural evolution in the growth process of FePt embedded in MgO matrix, J. Mater. Sci. (2020) 55:12305–12313.
- Jian Yu, Tingting Xiao, Xuemin Wang, Xiuwen Zhou, Xinming Wang, Liping Peng, Yan Zhao, Jin Wang, Jie Chen, Hongbu Yin, and Weidong Wu, A Controllability Investigation of Magnetic Properties for FePt Alloy Nanocomposite Thin Films, Nanomaterials, 2019, 9(1): 53.
Moreover, the Authors should clearly show here in what sense their materials are better than the others investigated earlier. As a conclusion, the topic could be interesting, due to the above criticisms I cannot recommend this paper to be published by Nanomaterials - an Open Access Journal from MDPI.
Response 2:We must declare that the figures in this paper are totally different with those we published earlier. Firstly, the subjects of these papers are different. The paper published in “J. Mater. Sci. (2020)” studied the structural evolution in the growth process of FePt embedded in the MgO matrix. The paper published in “Nanomaterials, 2019” studied the influence of laser energy density on ordering parameter and magnetic properties of FePt nanocomposite. While this paper focus on the annealing condition effect on its mico-structure variation, phase transition and magnetic properties. We are working on the similar FePt structure system. Because the samples structure system is similar, of course some test patterns and figures look like each other. Secondly, the growth conditions of the samples from these papers are also different and they are not the same series of samples. As a conclusion, these papers are not similar and the figures are also different with each other.

Reviewer 3 Report
In this manuscript the authors present their results on the effect of annealing conditions on the phase transition A10 => L10 for FePt nanoparticles in MgO matrix. They show, in particular, the importance of operating in a reducing atmosphere to achieve the transition. Working under vacuum does not prevent the oxidation of Fe into ferrite, which pertubs the transition.
These results, which without being completely original, can be interesting to carry out the phase transition.
One can nevertheless regret a rather summary bibliography: there has already been a lot of work on the annealing of FePt nanoparticles, in matrix (especially NaCl ) or under UHV. No mention is made of them. Moreover, there are only 18 references that are far from covering the most important work in the field. Authors should show in which of them their work is original.
Other results are problematic:
Report on the manuscript titled « Annealing Condition Effects on the Structural Properties of FePt Nanoparticles Embedded in MgO via Pulsed Laser Deposition » by Xiao et al.
The nanoparticles are obtained by laser ablation and deposited on a substrate heated to 650°C, can there not be a pre-set order that would subsequently promote the transition. It would be interesting to have a HAADF-STEM characterization of the nanoparticles after synthesis (sample S1 and also S2 and S4 for comparison).
The MgO matrix is compact and very well crystallized, what is the mechanism that allows the oxidation of iron under primary vacuum? A diffusion of O2 through the matrix? this seems unlikely, do the authors have an explanation?
I think that these different points must be treated before publication.
Reviewer 4 Report
Comments on the work dealing with the ‘Annealing Condition Effects on the Structural Properties of FePt Nanoparticles Embedded in MgO via Pulsed Laser Deposition’
- Introduction
Maybe the introduction could be improved, please.
About the FePt alloy and its great research interest, only applications as ultrahigh density magnetic recording media can be found ?
The references [1-4] are published in 2000/2013/2007/2010 : There is no more recent works?
- Experimental details
‘Detailed growth information was reported in our previous works [11, 12].’
Maybe the authors could illustrate in this paper these details. And try to emphase what is new or not in this present paper.
The authors think that how the system develops may affect/dicdate the annealing behavior ?
- Results and discussion
Figure 1 & Figure 2. What is the scale ? The scale is difficult to read.
Figure 3 & Figure 4 compare only 800°C/4h, maybe notes it. Annealing conditions is only the Atmosphere, T is 800°C and Time is 4h.
Do the authors tried different annealing Time ? What can be the consequence ?
The authors can add, please, the analysis results with S3 to discuss about the temperature effect, and see if the temperature can significally affect or note what is expected.
And for a better discussion, a third temperature annealing and analysis could be proposed ?
Even under vacuum and H2/Ar ?
Idem in Figure 6, please the authors can enrich the discussion with comparison with T annealing under O2 for instance ? Even under vacuum and H2/Ar ?
Author Response
Point 1:
1. Introduction
Maybe the introduction could be improved, please.
About the FePt alloy and its great research interest, only applications as ultrahigh density magnetic recording media can be found ?
The references [1-4] are published in 2000/2013/2007/2010 : There is no more recent works?
Response 1:Thanks for your great suggestions. We have modified the introduction and add some recent works and other applications, such as biological imaging and enhanced catalyst for oxygen reduction reaction.
Point 2:
2. Experimental details
‘Detailed growth information was reported in our previous works [11, 12].’
Maybe the authors could illustrate in this paper these details. And try to emphasis what is new or not in this present paper.
Response 2:Thanks for your great suggestions. We have already added all the useful growth information.
Point 3:
The authors think that how the system develops may affect/dicdate the annealing behavior ?
Response 3: We still don’t know how to improve the system for the moment, but we think the best way is to anneal FePt in a high purity reducing atmosphere in order to eliminate the influence of oxygen.
Point 4:
3. Results and discussion
Figure 1 & Figure 2. What is the scale ? The scale is difficult to read.
Response 4:We are very sorry for our negligence. We have modified the scale in Fig.1.
Point 5:
Figure 3 & Figure 4 compare only 800°C/4h, maybe notes it. Annealing conditions is only the Atmosphere, T is 800°C and Time is 4h.
Response 5:Thanks for your great suggestions. We have added the annealing temperature and time information in modified manuscript.
Point 6:
Do the authors tried different annealing Time? What can be the consequence?
Response 6:We only tried different annealing time in vacuum (1 hour and 2 hours, the longest time is 4 hours), but they both show disordered structure. We didn’t anneal with different time in H2/Ar yet, and we can do that later. Kim (Dispersible Ferromagnetic FePt Nanoparticles, Adv. Mater. 2009, 21, 906-909) report that the ordered transition is much more difficult because of the robust MgO coating. It requires longer annealing time (they spend 6 hours in H2/Ar). So, we may try longer annealing time. As for the situation of annealing in oxygen, we think longer annealing time will result severe oxidation.
Point 7:
The authors can add, please, the analysis results with S3 to discuss about the temperature effect, and see if the temperature can significally affect or note what is expected. And for a better discussion, a third temperature annealing and analysis could be proposed ?Even under vacuum and H2/Ar ?
Response 7:Thanks for your great suggestions. But in this work, we focus on annealing atmosphere (unannealed, annealed in oxygen, H2/Ar and vacuum) influence on its ordering process and try to figure out why they are not ordered when annealed in vacuum and oxygen. Further study about temperature influence and time influence on its oxidation process will be conducted in our next work.Your suggestion is a good idea and is also meaningful. As the situation annealed in H2/Ar, higher annealing temperature in H2/Ar might suppress or even eliminate iron oxide. C.W. White (J. Appl. Phys., 95 (2004) 8160.) report serious oxidation in UHV (2×108 Torr), flowing O2 or Ar (~99.999% pure). But the sample annealed under H2/Ar at 1100 degree shows no Fe oxide.
Point 8:
Idem in Figure 6, please the authors can enrich the discussion with comparison with T annealing under O2 for instance? Even under vacuum and H2/Ar ?
Response 8:We already test and added the magnetization curves of S3 (700 degree, annealing under O2). It also show soft ferromagnetic. Further study on annealing temperature and time influence on its oxidation process will be conduct in our future work.

Round 2
Reviewer 3 Report
This revised manuscript is now sufficiently improved for publication.